# An Overview of the Mechanisms through Which Plants Regulate ROS Homeostasis under Cadmium Stress

**DOI:** 10.3390/antiox13101174

**Published:** 2024-09-26

**Authors:** Pan Luo, Jingjing Wu, Ting-Ting Li, Peihua Shi, Qi Ma, Dong-Wei Di

**Affiliations:** 1College of Life Science and Technology, Gansu Agricultural University, Lanzhou 730070, China; luopan@gsau.edu.cn; 2Institute of Food Crops, Jiangsu Academy of Agricultural Sciences, Nanjing 210014, China; jjwu@jaas.ac.cn; 3State Key Laboratory of Soil and Sustainable Agriculture, Institute of Soil Science, Chinese Academy of Sciences, Nanjing 211135, China; litingting@issas.ac.cn; 4University of Chinese Academy of Sciences, Beijing 100049, China; 5University of Chinese Academy of Sciences, Nanjing (UCASNJ), Nanjing 211135, China; 6Department of Agronomy and Horticulture, Jiangsu Vocational College of Agriculture and Forestry, Jurong 212400, China; shipeihua@jsafc.edu.cn

**Keywords:** cadmium, antioxidant enzymes, antioxidants, reactive oxygen species, regulatory mechanism

## Abstract

Cadmium (Cd^2+^) is a non-essential and highly toxic element to all organic life forms, including plants and humans. In response to Cd stress, plants have evolved multiple protective mechanisms, such as Cd^2+^ chelation, vesicle sequestration, the regulation of Cd^2+^ uptake, and enhanced antioxidant defenses. When Cd^2+^ accumulates in plants to a certain level, it triggers a burst of reactive oxygen species (ROS), leading to chlorosis, growth retardation, and potentially death. To counteract this, plants utilize a complex network of enzymatic and non-enzymatic antioxidant systems to manage ROS and protect cells from oxidative damage. This review systematically summarizes how various elements, including nitrogen, phosphorus, calcium, iron, and zinc, as well as phytohormones such as abscisic acid, auxin, brassinosteroids, and ethylene, and signaling molecules like nitric oxide, hydrogen peroxide, and hydrogen sulfide, regulate the antioxidant system under Cd stress. Furthermore, it explores the mechanisms by which exogenous regulators can enhance the antioxidant capacity and mitigate Cd toxicity.

## 1. Introduction

Cadmium (Cd) is a water-soluble, non-essential element that is highly toxic to nearly all living organisms due to its neurotoxic and mutagenic properties [1]. Cd^2+^ is readily absorbed by plant roots and translocated to shoots and grains, even at low soil concentrations, thereby entering the food chain and posing a toxicity risk to humans [2,3]. Human activities such as the application of phosphate fertilizers, and the use of pesticides and herbicides further exacerbate Cd contamination in agricultural soils [4]. In plants, elevated Cd levels accelerate the production of reactive oxygen species (ROS) and lipid peroxides, leading to oxidative stress [5]. Additionally, Cd^2+^ accumulation inhibits photosynthesis and reduces water and nutrient uptake [6], resulting in growth retardation and, ultimately, plant death [7].

To survive under Cd stress, plants have developed various strategies to mitigate Cd toxicity, including the following: (i) Plants release specific root exudates that limit Cd^2+^ uptake from the soil. For example, jimsonweed (*Datura stramonium*) secretes lubimin and 3-hydroxylubimin, sorghum (*Sorghum bicolor*) releases malate, and tomato (*Solanum lycopersicum*) plants secrete oxalate, all of which reduce the Cd^2+^ uptake from Cd-contaminated soils [8,9,10]. (ii) Plants can also regulate Cd^2+^ uptake by modulating the expression levels of Cd^2+^ transporter genes [11,12]. However, no Cd^2+^-specific transporters have been identified in the plasma membranes of root cells to date. Instead, plants typically absorb Cd^2+^ through transporters that facilitate the uptake of other divalent ions, such as zinc (Zn^2+^), iron (Fe^2+^), calcium (Ca^2+^), manganese (Mn^2+^), magnesium (Mg^2+^), and copper (Cu^2+^) [11]. Several transporters with a Cd transport capacity have been identified, including zinc/iron-regulated transporter-like proteins (ZIPs), natural resistance-associated macrophage proteins (NRAMPs), heavy metal ATPases (AtHMA4 and AtHMA9), cation exchanger family members (CAX2 and CAX5), and ABC transporters (OsPDR9 and AtPDR8) [13,14,15,16,17]. (iii) When intracellular Cd^2+^ reaches a certain concentration, plants synthesize small metal-binding peptides called phytochelatins (PCs) and cysteine-rich, metal-binding proteins known as metallothioneins (MTs). These molecules chelate Cd^2+^, forming various Cd-PC and Cd-MT complexes, which are then sequestered in the vacuole by vacuolar transporters [18,19]. The role of PCs and MTs in detoxifying Cd toxicity has been demonstrated in several species, including bacopa (*Bacopa monnieri*), nigrum (*Solanum nigrum*), and moringa (*Moringa oleifera*) [20,21,22]. (iv) Finally, plant cells have developed a sophisticated antioxidant defense system to counteract the harmful effects of ROS bursts induced by Cd^2+^ accumulation [23]. The antioxidant system consists of enzymatic antioxidants, including superoxide dismutase (SOD), catalase (CAT), ascorbate peroxidase (APX), peroxidase (POD), guaiacol peroxidase (GPX), glutathione reductase (GR), dehydroascorbate reductase (DHAR), and monodehydroascorbate reductase (MDHAR), as well as non-enzymatic antioxidants, such as ascorbic acid (AsA), glutathione (GSH), carotenoids, proline, flavonoids, and phenolic compounds [18,24,25].

Although there have been numerous reviews summarizing the mechanisms of plant resistance to Cd stress, a systematic overview of the regulatory mechanisms of the antioxidant enzyme system and potential interactions remains lacking [19,23,26,27,28,29,30]. This review will provide a detailed summary of how Cd stress impacts the antioxidant system, how various elements, phytohormones, and signaling molecules regulate this system, the transcriptional and post-transcriptional regulation of the antioxidant system, and the role of exogenous modulators in enhancing antioxidant capacity under Cd stress.

## 2. Antioxidant Systems Involved in Regulating Cd-Induced ROS Bursts

Cd^2+^ accumulation in plants induces the production of ROS [31,32,33]. Under moderate stress conditions, plants can respond by enhancing their antioxidant systems to maintain intracellular redox homeostasis. However, excessively high concentrations of Cd^2+^ can impair the antioxidant system, leading to ROS accumulation and subsequent oxidative damage in plants [34]. The plant antioxidant system primarily comprises antioxidant enzymes, including SOD, CAT, and APX, as well as non-enzymatic antioxidants such as GSH and AsA (Figure 1) [4,35,36]. Numerous studies have demonstrated that the activity levels of antioxidant enzymes and non-enzymatic antioxidants vary in response to multiple factors, including plant species, plant organs, Cd^2+^ concentration, and the composition of the growth substrate [37,38,39,40].

Briefly, at the onset of oxidative scavenging, Cd-stress-induced O_2_^−^ is efficiently decomposed into H_2_O_2_ by SOD. Subsequently, H_2_O_2_ is reduced to H_2_O by APX or GPX within the ascorbic acid–glutathione (AsA-GSH) cycle, as well as by CAT and POD [4,41,42,43]. Two other key antioxidant enzymes, MDHAR and DHAR, are involved in the regeneration of AsA in cells, while GR is responsible for reducing oxidized glutathione back to GSH (Figure 1) [41,43]. Additionally, redox enzymes such as thioredoxin reductase and ferredoxin-thioredoxin reductase, along with non-enzymatic antioxidants like carotenoids, proline, and flavonoids, also play roles in degrading Cd-induced ROS [18,24,25,44,45].

When Cd^2+^ enters the cell, it can trigger an explosion of ROS. Both Cd^2+^ and ROS can directly activate the transcription of stress-responsive genes, including those encoding antioxidant enzymes and MTs, through specific transcription factors, thereby mitigating the damage caused by cadmium stress. On the one hand, the cell promotes the accumulation of MTs and PCs, enhancing the formation of Cd^2+^-MT and Cd^2+^-PC complexes, which are subsequently transported to the vacuole. This process reduces Cd^2+^ accumulation in the cytoplasm and decreases ROS production. On the other hand, the cell employs its antioxidant system, comprising both enzymatic and non-enzymatic components, to degrade excess ROS and alleviate oxidative stress.

## 3. Mechanisms by Which Different Mineral Elements Regulate ROS Homeostasis under Cd Stress

The balance of nutrient elements is critical for plant growth and development. However, Cd stress inhibits the uptake, translocation, and utilization of essential elements such as iron (Fe), zinc (Zn), calcium (Ca), phosphorus (P), and potassium (K). This disruption directly affects the integrity or activity of antioxidant enzymes that rely on these metal elements as cofactors, indirectly exacerbating oxidative stress [46]. Therefore, the appropriate supplementation of essential mineral elements can effectively alleviate Cd toxicity. The roles of different elements in regulating Cd toxicity are summarized below.

### 3.1. Nitrogen (N)

Nitrogen (N) is a macronutrient that is essential for plant growth and development [42]. Previous studies have shown that proper N application can alleviate Cd toxicity in plants. Interestingly, the form of the N significantly influences plant responses to Cd stress [47]. Nitrate (NO_3_^−^) and ammonium (NH_4_^+^) are the primary inorganic forms of N absorbed by plants [48,49]. Research indicates that the form of N can modulate the tolerance to Cd stress through the antioxidant system. For instance, in tomato plants, the availability of NH_4_^+^ was associated with the maintenance or even enhancement of key antioxidant enzymes such as CAT and APX under Cd stress [50]. In rice (*Oryza sativa*), an increase in the NH_4_^+^/NO_3_^−^ ratio significantly up-regulated the activities of key antioxidant enzymes, including SOD, CAT, and APX, as well as increasing concentrations of non-enzymatic antioxidants like AsA and GSH. While our latest research findings indicate that, under Cd stress, NH_4_^+^ promotes the accumulation of abscisic acid (ABA) more effectively than NO_3_^−^. This process activates OsbZIP20, which directly stimulates the transcription of *OsCATA* and *OsAPX2*, leading to enhanced activities of CAT and APX enzymes [51]. In summary, these results indicate that supplying NH_4_^+^ to plants can reduce Cd-induced ROS accumulation and enhance Cd tolerance [25,52,53]. However, opposite results were also observed in poplar (*Populus deltoids*) (Figure 2) [31]. Thus, the regulation of Cd tolerance by the N form varies between species, highlighting the need to investigate the molecular mechanisms of the N form regulation of Cd tolerance, rather than focusing solely on physiological aspects. Additionally, N forms also influence Cd^2+^ levels in plants (Figure 3) [31,54]. Therefore, selecting appropriate N fertilizers is crucial for improving Cd tolerance and reducing Cd^2+^ uptake in crops, thereby decreasing the risk of Cd^2+^ entering the food chain.

### 3.2. Phosphate (P)

Phosphorus (P) is another essential macronutrient involved in various cellular processes, including nucleic acid synthesis, membrane production, energy storage, and protein modification [55]. The impact of P fertilization on Cd^2+^ immobilization in soils has been widely studied in recent years, but there is limited research on the physiological functions of P in plants under Cd stress conditions [56,57]. Under Cd stress, P application in wheat (*Triticum aestivum*) was found to reduce Cd^2+^ uptake and slightly increase the uptake of K^+^, Ca^2+^, and Mg^2+^, while also enhancing the activities of antioxidant enzymes such as SOD, POD, and CAT (Figure 2 and Figure 3) [58]. However, opposite effects were observed in rice, where P application altered the subcellular distribution of Cd^2+^, leading to an increase in total Cd^2+^ content in roots and stems by enhancing the Cd^2+^ accumulation in the cell wall and reducing its presence in other cellular fractions [59,60]. Therefore, the mitigating effect of P on Cd toxicity appears to be closely related to both crop species and their specific environmental conditions, and the underlying molecular mechanisms require further investigation.

### 3.3. Calcium (Ca)

Ca is an essential macronutrient for plant growth and development, playing critical roles in regulating cell membrane and cell wall stability, ion transport, photosynthesis, enzyme activation, and acting as a second messenger [61]. Due to the similar ionic radii of Ca^2+^ and Cd^2+^, early studies on Ca’s role in mitigating Cd toxicity primarily focused on how Ca inhibits the uptake and transport of Cd^2+^ [62,63]. However, recent studies in wheat suggest that the inhibition of Cd^2+^ uptake by Ca^2+^ is influenced by Ca^2+^-associated anions and pH changes, with Ca^2+^ itself not directly reducing Cd^2+^ uptake but inhibiting its translocation from roots to shoots [64]. Additionally, Ca deficiency in rice has been shown to exacerbate Cd toxicity by increasing Cd^2+^ accumulation, inhibiting CAT activity, and reducing GSH levels, highlighting Ca’s potential role in regulating Cd-induced oxidative stress [65]. Applications of CaCl_2_, calcium polypeptide (CaP), and CaO nanoparticles (CaO-NPs) have been found to enhance Cd tolerance in rice by increasing the AsA content and boosting the activities of enzymes such as SOD, CAT, APX, GST, MDHAR, and DHAR [66,67,68]. However, the precise regulatory mechanisms remain unclear (Figure 2). Unlike other elements, research on how Ca^2+^ mitigates Cd toxicity must first determine whether Ca^2+^ functions solely as a mineral element or also as a second messenger.

### 3.4. Iron (Fe)

Fe is an essential micronutrient that acts as a cofactor in many enzymes and plays a crucial role in various metabolic processes, including photosynthesis, respiration, and DNA synthesis [69,70]. Cd stress often leads to Fe deficiency, but exogenous Fe application can mitigate this in the following two ways: by promoting the formation of Fe plaques that inhibit Cd^2+^ uptake, and by enhancing the plant’s resistance to Cd-induced oxidative stress through strengthening the antioxidant system (Figure 2 and Figure 3) [12,71]. Foliar application of 50 or 100 µM of Fe-lysine (an Fe-amino acid chelate) in canola (*Brassica napus*) and peas (*Pisum sativum*) significantly increases the activities of SOD, CAT, and POD enzymes, thereby reducing the accumulation of H_2_O_2_ and MDA induced by cadmium stress [69,72]. Similarly, foliar applications of Fe-NPs and Fe_3_O_4_-NPs significantly increased the activities of CAT, SOD, and APX in radishes (*Raphanus sativus*) and maize (*Zea mays*), while also reducing Cd-induced MDA accumulation [73,74]. However, the exogenous application of Fe-EDTA and FeCl_3_ in rice inhibited CAT, POD, and SOD activities and reduced MDA content, possibly due to Fe’s inhibition of Cd^2+^ uptake, which lessens stress and consequently the induction of the antioxidant system [75,76,77]. In conclusion, Fe supplementation through the root system promotes Fe plaque formation, while foliar Fe application strengthens the antioxidant system and enhances the scavenging of Cd-induced ROS. Proper Fe application is a reliable strategy to reduce Cd toxicity. However, it is important to note that although they reduce the overall Cd^2+^ content in the plant, high levels of Fe supplied to the roots carry the risk of increasing the Cd^2+^ accumulation in aboveground tissues [76].

### 3.5. Zinc (Zn)

Zn is an essential micronutrient for plants, playing a key role in various aspects of plant growth and metabolism, including photosynthesis, cell division, and maintaining membrane integrity [78]. As a cofactor for over 300 enzymes, Zn is crucial for both enzyme activity and structural stability [79]. Additionally, due to the similar physical and chemical properties of Zn and Cd, exogenous Zn has been investigated as a potential antagonist to Cd toxicity [80]. Indeed, several studies on wheat and rice have demonstrated that the exogenous application of Zn alleviates Cd-induced oxidative stress by enhancing the activities of key antioxidant enzymes, including SOD, POD, CAT, and APX [81,82]. Notably, ZnO nanoparticles (ZnO-NPs) have emerged as a significant tool in mitigating Cd toxicity in recent years [83]. Numerous studies have shown that ZnO-NPs can counteract Cd-induced ROS generation by upregulating the transcription of antioxidant enzyme genes and enhancing their activities. This effect has been observed in various species, including broccoli (*Brassica oleracea*), lettuce (*Lactuca sativa*), cabbage (*Brassica parachinensis*), maize, mung bean (*Vigna radiata*), rice, wheat, pepper (*Capsicum annuum*), and beans (*Vicia faba*) [84,85,86,87,88,89,90,91,92,93,94]. Additionally, ZnO-NPs have been found to reduce Cd^2+^ uptake and transport, thereby decreasing Cd^2+^ accumulation and toxicity in plants [83,95,96]. Interestingly, ZnO-NPs also promote the accumulation of salicylic acid (SA), and the combined action of SA and ZnO-NPs significantly enhances the antioxidant capacity of plants, suggesting a potential SA-dependent pathway through which ZnO-NPs exert their effects (Figure 2 and Figure 3) [83,97,98]. Given these promising findings, ZnO-NPs have garnered significant attention from researchers [83]. However, due to the distinct biological and physicochemical properties of ZnO-NPs, along with their potential risks to humans, such as ingestion, inhalation, and systemic diffusion via the circulatory system, further studies on the risk assessment of ZnO-NPs are necessary [99,100]. Addressing these risks is crucial to ensure the safe application of ZnO-NPs in agricultural practices [101,102].

### 3.6. Selenium (Se)

Se is an essential micronutrient for humans and animals, and it also plays a vital role in the growth and development of plants, which can benefit from moderate amounts of Se [103,104]. Se serves as a cofactor for several antioxidant enzymes, including SOD, GSH-Px, and CAT, enabling it to antagonize oxidative stress [27]. Numerous studies have demonstrated that the exogenous application of Se^2+^ or Se nanoparticles (Se-NPs) significantly enhances the antioxidant capacity of plants under Cd stress. This enhancement includes increased activities of antioxidant enzymes such as SOD, CAT, POD, APX, and GR, as well as elevated levels of GSH, leading to the reduced accumulation of H_2_O_2_ and MDA (Figure 2) [27,105,106,107,108,109,110,111,112,113,114,115]. Additionally, Se has been shown to significantly reduce Cd^2+^ uptake, thereby mitigating Cd stress (Figure 3) [27,105,107,110,112,113,115,116,117,118,119,120]. Particularly noteworthy is Se’s ability to respond to Cd stress by regulating phytohormones and other signaling molecules. For example, Se significantly promotes the accumulation of auxins and polyamines (PAs), while inhibiting the biosynthesis of ethylene (ET) and nitric oxide (NO), thus enhancing the Cd resistance of plants [108,115,121]. However, it is important to note that excessive Se can also exert stress on plants [117]. Therefore, when using Se to ameliorate Cd toxicity, careful attention must be paid to the dosage and method of application.

### 3.7. Silicon (Si)

Si is the second most abundant element in the Earth’s crust, following oxygen [122,123]. Although not essential for plant growth factors, Si is a beneficial element that plays a crucial role in supporting plant survival under heavy metal stress, including Cd stress [122,123,124]. Numerous studies have shown that the exogenous application of SiO_3_^2−^, Si nanoparticles (Si-NPs), and SiO_2_ nanoparticles (SiO_2_-NPs) can significantly enhance the activities of antioxidant enzymes, such as SOD, CAT, POD, GR, and AOX, and maintain GSH-AsA homeostasis, thereby improving a plant’s resistance to Cd-induced oxidative stress (Figure 2) [109,125,126,127,128,129,130,131,132,133,134,135,136,137]. Moreover, Si can regulate hormonal homeostasis by promoting the biosynthesis of brassinosteroids (BRs), jasmonic acid (JA), abscisic acid (ABA), and PAs, while inhibiting the accumulation of SA. It also synergizes with strigolactone (SL), BRs, and melatonin (MLT) to co-regulate the activities of enzymes such as SOD, CAT, and POD, thereby enhancing Cd tolerance in plants (Figure 2) [26,134,136,138,139,140]. Additionally, Si application has been reported to increase the uptake of mineral elements such as Mg, Fe, Zn, and Se, promote the formation of Fe plaques, and synergize with mineral elements such as K, Zn, Se, and boron to directly or indirectly modulate Cd^2+^ uptake and enhance antioxidant capacity (Figure 2 and Figure 3) [82,109,141,142,143,144].

These studies demonstrate that Si can enhance plants’ antioxidant capacity and improve their Cd resistance, with the added advantage that excess Si is not toxic to plant growth, making Si a potentially excellent amendment for mitigating Cd toxicity [134]. However, the specific molecular mechanisms underlying the Si-mediated reduction of Cd toxicity in plants remain poorly understood, and the practical application of different Si forms in soils requires further exploration and investigation.

### 3.8. Sulfur (S)

S is an essential macronutrient for plant growth and development, serving as a constituent of vitamins, Fe-S clusters, and defense compounds such as GSH, PCs, and hydrogen sulfide (H_2_S), all of which play critical roles in counteracting environmental stresses [145,146,147]. S can enhance plant resistance to Cd-induced oxidative stress by increasing the activities of antioxidant enzymes, including SOD, CAT, APX, GR, and DHAR, and by boosting levels of AsA and GSH [148,149,150,151]. Additionally, the application of S fertilizers increases H_2_S accumulation, which in turn activates H_2_S-dependent pathways that further enhance the antioxidant capacity of plants (Figure 2) [149]. Moreover, S regulates Cd^2+^ absorption and translocation in three key ways: by inducing the formation of CdS precipitates, thereby altering the bioavailability of Cd^2+^ in the soil [152,153]; by increasing the formation of Fe plaques on root surfaces, which reduces Cd^2+^ uptake [152,153]; and by promoting the chelation of Cd^2+^ by PCs, thereby controlling Cd translocation from roots to shoots [150,151,154,155]. However, the regulation of Cd^2+^ uptake and transport by S is closely linked to factors such as soil oxidation/reduction potential (Eh), the form of the S, and the crop species [148,150,151,154,155,156,157,158,159]. Therefore, careful consideration is required when applying S-containing fertilizers to Cd-contaminated soils.

### 3.9. Other Minerals

Cerium (Ce), a rare earth element, has been reported to enhance Cd resistance by increasing the activity of antioxidant enzymes such as SOD, POD, and CAT in rice (Figure 2) [160]. However, a recent study indicated that applying 500 mg kg^−1^ of CeSO_4_ led to higher Cd^2+^ accumulation in both the roots and shoots of maize seedlings, while the same concentration of CeO_2_ nanoparticles (50 nm) promoted maize growth under Cd stress. This suggests that the mitigating effect of Ce on Cd stress is closely related to both the form and dosage of Ce [161]. Boron (B) has also been shown to increase the activities of CAT, SOD, and POD, while inhibiting Cd^2+^ uptake and transport, thereby improving Cd resistance in rice (Figure 2) [162]. Additionally, under Cd stress, arsenic (As) has been found to significantly reduce Cd^2+^ accumulation in rice and alleviate oxidative stress by increasing POD and SOD activities, as well as GSH and AsA levels (Figure 2) [163]. Moreover, TiO_2_ nanoparticles (TiO_2_-NPs) have been reported to increase the activities of SOD, CAT, POD, and APX, enhancing the resistance of maize and rice to Cd-induced oxidative stress. This effect is partly achieved by inducing the expression of genes related to Fe uptake and utilization in plants (Figure 2 and Figure 3) [131,164].

The above studies demonstrate that the application of mineral nutrients can reduce Cd^2+^ accumulation and enhance antioxidant capacity. However, further research is needed to investigate the molecular mechanisms through which different mineral elements mitigate Cd toxicity, as well as the roles of hormones and signaling substances. Additionally, the relationship between different combinations of mineral elements and/or phytohormones (signaling molecules) and Cd tolerance warrants further exploration.

## 4. The Role of Plant Growth Regulators and Signaling Molecules in Regulating ROS Homeostasis under Cd Stress

Plant hormones, the most well-known growth regulators, including ABA, auxin, ET, BR, JA, and gibberellins (GAs), play a crucial role in regulating plant growth and development, as well as in the response to environmental stress. However, Cd stress disrupts the balance of endogenous hormone levels. As a result, the exogenous application of appropriate concentrations of phytohormones or their inhibitors can help alleviate Cd toxicity in plants [25,165]. In addition to phytohormones, plants utilize various other growth regulators and signaling molecules, such as MLT, SA, H_2_S, NO, and H_2_O_2_, to manage cadmium (Cd) stress.

### 4.1. Plant Growth Regulators

#### 4.1.1. Abscisic Acid (ABA)

ABA is widely recognized as a “stress hormone” involved in the regulation of various environmental stresses, including salinity, drought, and heavy metal exposure [166,167,168,169]. Numerous studies have shown that Cd stress induces ABA accumulation, which enhances Cd resistance by boosting antioxidant enzyme activities and increasing antioxidant levels to scavenge Cd-induced ROS (Figure 2) [170]. Research has demonstrated that ROS accumulation increases and Cd resistance decreases in tomato ABA-deficient mutants under Cd stress [171]. Conversely, the exogenous application of ABA has been shown to mitigate Cd-induced ROS accumulation by enhancing the activities of antioxidant enzymes such as SOD, POD, GR, CAT, and APX, as well as by increasing the levels of antioxidants like proline and AsA in various plant species, including lettuce and mung bean [172,173,174,175,176]. Although several transcription factors (TFs) have been identified as being involved in ABA’s regulation of antioxidant enzymes, the precise molecular mechanisms remain unclear [177]. Until recently, we discovered that OsbZIP20 is induced by Cd in rice and enhances its antioxidant capacity by directly activating *OsAPX2* and *OsCATA* [51]. Beyond enhancing antioxidant capacity under Cd stress, ABA has also been reported to improve Cd tolerance in plants by reducing Cd^2+^ uptake and translocation, as highlighted in previous reviews (Figure 3).

#### 4.1.2. Auxin

Auxin is a critical phytohormone involved in regulating plant growth, development, and stress responses [178]. Studies have shown that Cd exposure promotes root hair development by stimulating ROS bursts, a process in which auxin plays a significant role. Additionally, the exogenous application of auxin biosynthesis inhibitors, such as yucasin and 4-phenoxyphenylboronic acid, has been found to significantly reduce ROS bursts in tomato, rice, and Arabidopsis under Cd stress [179,180,181,182]. Furthermore, auxin has been shown to cause the denitrosylation of AtAPX1, partially inhibiting APX1 activity in Arabidopsis roots, suggesting that auxin acts as a facilitator in Cd-induced ROS bursts [183].

Several studies have also indicated that auxin can influence Cd stress by regulating Cd^2+^ accumulation, though the findings are conflicting. For instance, the exogenous application of the auxin, NAA, has been reported to increase the hemicellulose content in the cell walls of Arabidopsis, thereby enhancing Cd^2+^ fixation in the cell wall, reducing Cd^2+^ transport to aboveground parts, and decreasing Cd toxicity. Conversely, in rice, NAA has been shown to decrease the hemicellulose content in the cell wall and reduce Cd^2+^ uptake, resulting in increased Cd tolerance [184,185].

Moreover, there are differing conclusions on how Cd regulates auxin levels [108,180,184,185,186]. These results suggest a complex role for auxin in response to Cd stress [180]. The mechanism of auxin’s response to Cd stress may depend on factors such as the species, the site of action, and the timing, necessitating further in-depth studies.

#### 4.1.3. Brassinosteroids (BRs)

BRs are phytohormones widely involved in the regulation of plant growth and development [32]. BRs also play a significant role in responding to various abiotic stresses, including drought, salinity, and heavy metal stress [187,188,189]. Studies have shown that the exogenous application of BRs, such as 28-homobrassinolide or 24-epibrassinolide, enhances the activities of SOD, CAT, and APX, and increases proline content, thereby protecting tomato plants from Cd-induced oxidative stress. This suggests that BRs contribute to enhancing Cd tolerance in plants (Figure 2) [189,190]. It is proposed that BRs may function by inhibiting and/or de-repressing the transcription of specific genes, inducing protein synthesis, and activating enzyme activity [189,191,192]. However, there remains a gap in the research regarding the regulation of the antioxidant system by BRs under Cd stress.

#### 4.1.4. Ethylene (ET)

ET is another well-known “stress hormone” that plays a crucial role in defense responses to a wide range of stresses, including both biotic and abiotic factors [193,194]. Several studies in sunflowers, peas, and wheat have shown that Cd stress induces ET accumulation by upregulating the transcription of *ACS* genes [195,196,197,198]. It has been observed that Cd stress promotes root hair growth by enhancing ET biosynthesis, which in turn increases ROS accumulation in Arabidopsis root hairs [182]. As the Cd^2+^ concentration increases beyond 40 µM, both ET and ROS levels decrease compared to 40 µM of Cd^2+^, but they remain higher than in the control (0 µM Cd), suggesting that the Cd regulation of ET and ROS content is concentration dependent, at least in root hairs [182]. Interestingly, another study found that mild Cd stress (5 µM) rather than higher concentrations (e.g., 10 µM) promoted ROS production through the activation of alternative oxidase 1a (AOX1a), a terminal oxidase of the plant mitochondrial electron transport chain involved in regulating mitochondrial ROS production, potentially involving ET and NO [199,200,201]. These findings suggest that ET may function as a downstream signaling pathway in Cd-induced ROS production, and that this signaling pathway is dependent on the Cd concentration.

#### 4.1.5. Jasmonic Acid (JA)

The role of JAs in controlling plant tolerance to abiotic stresses such as salinity, drought, temperature, and heavy metals has been extensively studied [202]. Several studies have shown that Cd stress promotes the accumulation of endogenous JA [203,204,205,206]. Externally applied methyl jasmonate (MeJA) has been found to enhance the activities of antioxidant enzymes such as SOD, GPX, CAT, GR, APX, and POD, as well as increasing proline and GSH levels under Cd stress, thereby mitigating the oxidative damage induced by Cd (Figure 2) [203,205,206]. Furthermore, unlike other hormones, the regulation of APX enzyme activity by JA has been further elucidated at the protein structure level in rice. Cd^2+^ and AsA share similar binding sites on APX, and Cd^2+^ binding leads to changes in protein conformation, affecting the formation of the [APX-AsA] complex and reducing APX activity. In contrast, in the presence of JA, the interaction required for [OsAPX-AsA] complex formation is restored, helping to maintain APX activity under Cd stress [207]. Additionally, JA enhances Cd resistance by promoting ABA biosynthesis, reducing Cd^2+^ uptake, and increasing the uptake of other essential mineral nutrients (Figure 3) [203,206].

#### 4.1.6. Salicylic Acid (SA)

SA is a ubiquitous phenolic hormone widely recognized for its extensive involvement in local and systemic plant defense responses [208]. An increasing number of studies have shown that SA also plays a crucial role in enhancing tolerance to both biotic and abiotic stresses [209]. Under Cd stress, endogenous SA accumulation is promoted, while exogenous SA application has been found to reduce H_2_O_2_ and MDA accumulation, suggesting that SA can alleviate the oxidative damage caused by Cd stress (Figure 3) [210,211,212,213]. Further studies have revealed that SA enhances the activities of enzymes such as SOD, POD, DHAR, and GR under Cd stress, though its effect on CAT and APX activities has produced contradictory results (Figure 2) [210,211,212,213,214,215]. Some studies found that exogenous SA treatment specifically inhibited CAT and APX activities, while others reported an enhancement of these activities [209]. This discrepancy may be due to SA’s ability to chelate Fe, and its role as a single electron donor, potentially affecting the CAT-dependent peroxidation cycle [209,216]. The content of endogenous non-chelated Fe may play a key regulatory role in determining CAT and APX enzyme activities.

In addition to these roles, SA has unique functions: it can directly scavenge hydroxyl radicals as an antioxidant [211,214], and it can bind to Cd^2+^ to form a Cd-SA complex, thereby reducing Cd^2+^ translocation [211,214,217]. Moreover, SA enhances Cd resistance in plants by promoting ABA synthesis, decreasing Cd^2+^ uptake, and increasing mineral uptake under Cd stress [209,211,217,218].

#### 4.1.7. Melatonin (MLT)

MLT, also known as N-acetyl-5-methoxytryptamine, is a plant growth regulator that functions as a hormone in mammals. In plants, MLT functions as an antioxidant and growth regulator, playing roles in various physiological processes, including germination, photosynthesis, flowering, leaf senescence, root development, carbohydrate metabolism, and circadian rhythms, as well as in responses to environmental stresses [219,220]. Several studies have demonstrated that MLT enhances the resistance to Cd-induced oxidative stress by increasing the activities of antioxidant enzymes such as SOD, POD, CAT, and APX (Figure 2) [220,221]. Interestingly, MLT has been found to synergistically interact with H_2_S to inhibit Cd^2+^ uptake and enhance antioxidant enzyme activity, thereby reducing Cd toxicity [220]. Moreover, a study in pepper plants showed that MLT promotes H_2_S biosynthesis, which in turn activates the antioxidant system and enhances the resistance to arsenic (As). The mitigating effect of MLT on As toxicity was abolished by the addition of H_2_S scavengers such as hypotaurine, suggesting that MLT alleviates As toxicity through an H_2_S-dependent pathway [222]. This raises the question of whether MLT similarly responds to Cd stress through H_2_S-dependent or independent pathways, which warrants further investigation. Additionally, MLT is known to regulate fruit ripening by promoting the accumulation of ET and ABA. Exploring whether this regulatory effect also plays a role in the plant response to Cd stress is equally deserving of in-depth study [223].

#### 4.1.8. Other Phytohormones

In addition to the previously mentioned phytohormones, GA, strigolactone (SL), and PAs have also been reported to enhance Cd resistance by reducing Cd^2+^ uptake and increasing antioxidant capacity [121,224,225,226]. It has been observed that Cd stress can alter the levels of nearly all hormones, and the application of exogenous hormones or biosynthesis inhibitors can enhance plant responses to Cd stress by boosting their antioxidant capacity [23,25]. However, it remains unclear whether these changes in hormone levels are a direct response to Cd stress or a consequence of plant damage caused by Cd exposure. Elucidating these mechanisms will provide a solid theoretical foundation for a deeper understanding of plant responses to Cd stress and the enhancement of Cd resistance.

### 4.2. Signaling Molecules

#### 4.2.1. Nitric Oxide (NO)

In plants, NO is a transient gaseous signaling molecule that enhances tolerance to heavy metal stress by boosting the antioxidant defense system [227,228]. Under Cd stress, endogenous NO levels are reduced; however, the exogenous application of the NO donor sodium nitroprusside (SNP) significantly increases the activities of antioxidant enzymes such as SOD, CAT, POD, and APX, as well as proline levels, thereby reducing the accumulation of Cd-induced H_2_O_2_ and MDA in peas and wheat (Figure 2) [72,229,230,231]. Consistently, the overexpression of rat neuronal NO synthase in rice, which raises endogenous NO levels, was found to promote the transcription of *OsCATA*, *OsCATB*, and *OsPOX*, thereby enhancing their enzyme activities under Cd stress [232]. Additionally, NO promotes the biosynthesis of ABA and GA under Cd stress, activating ABA- and GA-dependent Cd tolerance mechanisms [233]. In summary, NO regulates antioxidant enzyme activity through five primary mechanisms: (i) activating the transcription of genes encoding antioxidant enzymes [232]; (ii) promoting Fe uptake, which is an essential cofactor for the functioning of CAT and POD [72,229,230,231,234]; (iii) directly interacting with ROS to prevent oxidative damage [235]; (iv) increasing endogenous ABA and GA accumulation under Cd stress [233]; and (v) accelerating GSH accumulation by directly upregulating the genes encoding rate-limiting enzymes in GSH biosynthesis, such as gamma-glutamylcysteine synthetase (γ-ECS) and glutathione synthetase (GSHS) [236].

However, numerous studies suggest that NO does not always mitigate Cd toxicity; in some cases, it can actually exacerbate Cd toxicity [237]. These studies indicate that Cd stress can promote NO synthesis, and NO accumulation may intensify Cd toxicity by inhibiting Cd^2+^ chelation by PCs and MTs, enhancing Cd^2+^ uptake, and reducing the activities of antioxidant enzymes such as CAT and APX (Figure 2) [72,237,238,239,240,241]. In fact, several factors influence the regulation of NO content under Cd stress, including the Cd^2+^ concentration, the treatment duration, and the plant species [237]. The mitigating effect of NO on Cd stress is closely related to its concentration; therefore, the dual effect of NO on Cd stress may depend on the level of endogenous NO accumulation [242,243,244]. Although it has been reported that nitric oxide synthase (NOS) and nitrate reductase (NR) are involved in the regulation of NO production under Cd stress, the specific mechanisms of Cd-induced NO production in plants remain unclear and require further investigation. Understanding these mechanisms is crucial for clarifying the role of NO in regulating Cd stress [237].

#### 4.2.2. Hydrogen Peroxide (H_2_O_2_)

H_2_O_2_ is not only a byproduct of oxidative stress and a free radical but also functions as a signaling molecule that regulates plant growth, development, and stress responses [245]. Several studies have demonstrated that the exogenous application of low concentrations of H_2_O_2_ can mitigate the toxic effects of Cd stress on plants, with the effective concentration depending on the plant species. For instance, 0.1 μM of H_2_O_2_ has been shown to protect tomato plants from Cd stress by enhancing CAT and APX activities and increasing the accumulation of AsA and GSH (Figure 2) [246]. In rapeseed, pretreatment with 50 μM of H_2_O_2_ significantly increased the GSH content, thereby enhancing Cd tolerance [41]. Additionally, 100 μM of H_2_O_2_ has been found to reduce Cd^2+^ uptake, enhance the activities of SOD, CAT, APX, GPX, GR, and GST, and increase GSH and PC levels, thereby reducing Cd toxicity in rice [247,248,249,250,251].

Indeed, H_2_O_2_ does not act independently as a fundamental signaling molecule in many signaling pathways; rather, it functions within dynamic systems, interacting with several other phytohormones (such as GAs, CKs, ABA, JA, ET, BR, and SA) and signaling molecules (including NO and Ca^2+^) [245]. Recent molecular studies have revealed that H_2_O_2_ is also involved in the modulation of MAPK- and TF-dependent signal transduction, which is a critical strategy for plants to cope with environmental stresses [246,252]. Therefore, an in-depth study of the molecular mechanisms underlying the H_2_O_2_ signaling pathway is particularly important for understanding how plants effectively respond to various environmental stresses, including heavy metal stress.

#### 4.2.3. Hydrogen Sulfide (H_2_S)

H_2_S, an emerging signaling molecule, plays a significant role in plant stress responses, including responses to heavy metal stress [253,254]. H_2_S mitigates Cd toxicity in plants through several mechanisms, including the induction of APX persulfation [255,256], the stimulation of the AsA-GSH cycle [257], the protection of mitochondrial function [258], and the enhancement of antioxidant enzyme activities [254]. A comprehensive meta-analysis demonstrated that exogenous H_2_S increased the activities of CAT (~39.51%), POD (~22.59%), APX (~17.80%), and SOD (~12.86%), thereby reducing Cd stress-induced H_2_O_2_ (~24.52%) and MDA (~20.37%) levels [259]. Moreover, the effectiveness of H_2_S in mitigating Cd toxicity is closely related to the plant species, the application method of NaHS (an H_2_S donor), and the growing conditions [259].

## 5. Transcriptional and Post-Transcriptional Modification Regulation of ROS Homeostasis under Cd Stress

Several TFs have been reported to respond to Cd stress by regulating the activities of antioxidant enzymes or the levels of non-enzymatic antioxidants (Table 1). For instance, the transcription of *AtMYB4*, *SaHsfA4c*, and *ZmWRKY4* is induced, while *AtWRKY12* is inhibited by Cd stress [177,260,261,262]. Additionally, these TFs have been implicated in regulating Cd^2+^ chelation, uptake, and transport, which consequently affects the Cd^2+^ content and distribution in plants [33,260,262,263,264]. Therefore, it remains to be verified whether these TFs directly regulate the transcription of antioxidant-related genes or indirectly influence the antioxidant capacity by altering the Cd^2+^ content in plants.

Beyond transcriptional regulation, post-transcriptional modifications also play a role in modulating antioxidant enzyme activities under Cd stress. For example, in pea leaves, Cd stress increases CAT activity and reduces H_2_O_2_ accumulation by maintaining the S-nitrosylation level of CAT [72]. Interestingly, APX can be both inactivated by irreversible nitration and activated by reversible S-nitrosylation under salt stress [265]. Additionally, CALCIUM-DEPENDENT PROTEIN KINASE 8 (CPK8) has been shown to phosphorylate CAT3 in an ABA-dependent manner, thereby activating its activity and regulating H_2_O_2_ homeostasis under drought stress in Arabidopsis [266]. However, whether these post-transcriptional modifications also occur under Cd stress remains to be further investigated.
antioxidants-13-01174-t001_Table 1Table 1Transcription factors involved in regulating Cd-induced oxidative stress.TF NamePlant SpeciesFunctions in PlantTarget GenesRef.AtMYB4*Arabidopsis thaliana*AtMYB4 positively regulates Cd tolerance via promoting PCs and MT accumulation, and enhancing CAT, SOD, and APX activities.*AtPCS1*; *AtMT1C*; *AtCAT3*; *AtSOD*; *AtAPX1*[260]AtMYB75*Arabidopsis thaliana*AtMYB75 positively regulates Cd tolerance via promoting anthocyanin, GSH, and PC biosynthesis and enhancing SOD, and CAT activities.*AtACBP2*; *AtABCC2*[33]AtWRKY12*Arabidopsis thaliana*AtWRKY12 negatively regulates Cd toxicity via decreasing GSH and PC contents.*AtGSH1*; *AtGSH2*; *AtPCS1*; *AtPCS2*[262]CaPF1*Capsicum annuum*Overexpression of CaPF1 increased Cd tolerance via enhancing APX, GR and SOD activities in Virginia pine.*None*[267]MsbHLH115*Medicago sativa*Overexpression of MsbHLH115 increased Cd tolerance via enhancing Fe/Zn accumulation and SOD, POD and CAT acitivities in Arabidopsis.*MsbHLH121*; *AtSOD1*; *AtPOD1*[264]PyWRKY75*Populus yunnanensis*Overexpression of PyWRKY75 increased Cd tolerance via enhancing POD, SOD, CAT, and APX activities, and increasing AsA, GSH, and PCs accumulation.None[263]SaHsfA4c*Sedum alfredii*Overexpression of SaHsfA4c increased Cd tolerance via enhancing CAT, APX, and POD activities in Arabiposis and Sedum alfredii.*SaCAT*; *SaAPX*; *SaPOD*[261]ThWRKY7*Tamarix hispida*ThWRKY7 can specifically bind to and activate ThVHAc1 and improve Cd stress tolerance by enhancing SOD, POD, and GPX acitivities.*ThVHAc1*; *ThSOD*; *ThGPX*; *ThPOD*[268]OsbZIP20*Oryza sativa*OsbZIP20 can specially bind to and activate the transcription of *OsCATA* and *OsAPX2* and improve Cd stress tolerance by enhancing CAT and APX activities.*OsCATA*; *OsAPX2*[51]ZmWRKY4*Zea mays*Overexpression of ZmWRKY4 positively regulates Cd tolerance via enhancing SOD and APX activities.*ZmSOD4*; *ZmAPX*[177]


## 6. Application of Amendments for Mitigating Cd-Induced Oxidative Stress

In addition to the application of essential mineral elements, phytohormones, and signaling molecules, the exogenous application of certain amendments can also mitigate Cd toxicity. The mechanisms by which these amendments operate can be summarized as follows: (i) reducing the bioavailability of Cd^2+^ in the soil, thereby limiting the amount of Cd^2+^ that plants can absorb; (ii) promoting the formation of Fe plaques on the root surface, which hinder Cd^2+^ uptake; (iii) increasing the contents of lignin, PCs, MTs, and other compounds to promote Cd^2+^ sequestration and reduce its translocation; and (iv) enhancing the plant’s antioxidant capacity. Interestingly, the enhancement of the antioxidant capacity by these amendments is often closely linked to the reduced bioavailability of Cd^2+^ within the plant.

### 6.1. Citric Acid (CA)

CA is an intermediate of the tricarboxylic acid (TCA) cycle that provides cells with energy for respiration and various biochemical activities [269]. Numerous studies have demonstrated that CA can enhance heavy metal remediation in plants by promoting metal solubility and mobilization, as well as by increasing the activity of key enzymes in the antioxidant defense system (Table 2) [270,271]. CA treatment has been shown to promote the uptake of Cd^2+^ by roots and to facilitate its translocation from roots to stems, thereby increasing the Cd^2+^ accumulation in plant roots and stems [271]. Additionally, CA treatment enhances the activities of enzymes such as POD, CAT, APX, DHAR, MDHAR, GR, and GPX, and helps maintain GSH-AsA homeostasis, thereby bolstering the antioxidant capacity of plants [1,271,272].

### 6.2. β-Aminobutyric Acid (BABA)

BABA is a natural plant metabolite that acts as a priming activator, inducing a “primed state” in plants, which provides broad-spectrum resistance to a wide range of stresses, including pathogen infections and various abiotic stresses [273,274,275]. Under Cd stress, BABA-pretreated plants have been shown to enhance Cd tolerance by increasing the activities of APX, POX, and MDAR enzymes and maintaining the regeneration capacity of the AsA-GSH cycle (Table 2) [273,276,277]. Studies in *Arabidopsis* further demonstrated that the exogenous addition of BSO, an inhibitor of GSH synthesis, eliminated BABA-induced Cd tolerance, suggesting that BABA may enhance Cd tolerance in *Arabidopsis* through a GSH-dependent pathway [273]. Additionally, research in strawberries showed that BABA pretreatment significantly increased the levels of endogenous NO and H_2_S while reducing ABA levels, thereby enhancing Cd tolerance [277]. This suggests that BABA, as a priming activator, may synergistically enhance the plant’s ability to tolerate Cd by prematurely inducing multiple stress tolerance pathways.

### 6.3. Nanoenzymes

Nanoenzymes are a class of artificial enzymes that have garnered attention for their ability to mimic the characteristics and catalytic properties of natural enzymes [274]. Calcium hexacyanoferrate nanozyme (CaHCF-NPs) is a novel type of nanoenzyme reported to ameliorate Cd toxicity by mimicking the catalytic properties of various natural enzymes such as SOD, POD, CAT, GPX, APX, and thiol peroxidase (TPX). The exogenous application of CaHCF has been shown to effectively decompose excessive reactive oxygen species (ROS), thereby reducing oxidative stress damage in plants (Table 2). Additionally, CaHCF nanoparticles can decrease the Cd^2+^ accumulation in plants through ion exchange with Cd^2+^, further mitigating the oxidative damage induced by heavy metals. Moreover, CaHCF nanoparticles have been found to upregulate the expression of genes related to antioxidant defense, heavy metal detoxification, nutrient transport, and stress resistance, thereby enhancing plant resilience [278]. Despite the promising role of CaHCF in enhancing Cd resistance, the residue and safety assessment of CaHCF nanoparticles remain insufficient. Continuous monitoring and analysis of CaHCF nanoparticles at different stages of crop development are necessary to ensure their safety.

### 6.4. Exopolysaccharides (EPSs)

EPSs are exogenous active substances secreted by microorganisms such as fungi, bacteria, and algae, which play a crucial role in plant–microbe interactions. EPSs can protect plant growth under abiotic stresses by maintaining a healthy rhizosphere environment [279,280]. Several studies have demonstrated that EPSs can effectively ameliorate Cd toxicity in rice (Table 2). On one hand, EPSs promote lignin synthesis and cell wall remodeling, enhancing Cd resistance. Additionally, EPSs inhibit the expression of genes such as *OsNramp1*, *OsNramp5*, and *OsHMA2*, reducing Cd^2+^ uptake and translocation. On the other hand, EPSs activate the oxidative stress response, leading to increased activities of SOD, CAT, POD, and the AsA-GSH cycle under Cd stress, thereby boosting the plant’s antioxidant capacity [43,281,282]. Moreover, EPSs have been shown to promote the synthesis of linolenic acid, a precursor of JA, suggesting that the JA signaling pathway may also be involved in the regulation of Cd stress resistance by EPS [283,284]. In addition to EPSs, microorganisms can promote plant growth and antioxidant activity by secreting other signaling molecules such as riboflavin and lumichrome. These microorganisms can also significantly increase the content of flavonoids and other phenolic compounds in plants, further enhancing their antioxidant capacity [34,271,285,286,287,288,289].
antioxidants-13-01174-t002_Table 2Table 2Amendments for enhancing antioxidant activity in plants under Cd stress.AmendmentsDosesCd^2+^MediumPlant SpeciesFunctions in PlantRef.CA0.5 mM/1.0 mM0.5 mM/1.0 mMSolution*Brassica juncea*Increasing the levels of AsA, GSH, and PC, while enhancing the activities of key antioxidant enzymes such as APX, MDHAR, DHAR GR, GPX, SOD, and CAT.[272]BABA 200 μM100 μM/150 μM1/2 MS*Arabidopsis thaliana*Increasing GSH levels.[273]20 mM//*Fragaria ananassa*Increasing the endogenous levels of NO, H_2_S, and AsA.[277]200 μM100 μMSolution*Glycine max*Priming and activating antioxidant defense mechanisms.[276]Nanoenzymes120 μg/mL50 μM 1/2 MS*Arabidopsis thaliana*Enhancing the activities of SOD, POD and CAT, while mimicking the activities of SOD, CAT, POD, TPX, GPX, and APX.[278]150 μM1/2 MS*Solanum lycopersicum*300 μMSolution 27 μg/mgSoilEPS100 mg/L/200 mg/L50 μMSolution*Oryza sativa*Enhancing the activities of CAT and POD, while increasing the levels of GSH.[281]250 mg/L30 μMSolution*Oryza sativa*Increasing GSH levels.[43]100 mg/L /400 mg/L10 μM/25 μMSolution*Oryza sativa*Promoting lignin biosynthesis and activating antioxidant activity.[282]NAC500 μM50 μM/200 μMSolution*Solanum nigrum*Enhancing the biosynthesis of GSH.[21]


### 6.5. N-Acetyl-Cysteine (NAC)

NAC is a compound derived from the acetylation of cysteine, which can be further degraded into cysteine and glutathione (GSH) in plants, acting as a broad-spectrum antioxidant. NAC has been shown to reduce the oxidative stress induced by herbicides and UV-B radiation [290,291,292]. A study in *Solanum nigrum* demonstrated that NAC treatment increased the activities of antioxidant enzymes such as SOD, POD, APX, GSH-Px, and GR in roots under Cd stress, thereby enhancing the Cd tolerance of nigrum (Table 2) [21]. This suggests that NAC not only functions as an antioxidant but also acts as an inducer, stimulating the activities of various antioxidants and thereby boosting the plant’s overall antioxidant capacity.

Additionally, materials such as sepiolite, bentonite, biochar, and certain microbes are also involved in enhancing antioxidant enzyme activities under Cd^2+^ stress [25,293,294].

## 7. Concluding Remarks

Cd is not a redox-active metal and, therefore, it cannot generate ROS via the Fenton or Haber–Weiss reactions. Instead, Cd enhances ROS accumulation by disrupting the electron transport chain, weakening the antioxidant system, or interfering with the uptake and metabolism of essential mineral nutrients [18,24]. Cd-induced ROS include singlet oxygen (^1^O_2_), superoxide anion (O_2_^−^), hydrogen peroxide (H_2_O_2_), and hydroxyl radicals (OH^−^), which oxidize cellular components such as proteins, lipids, and nucleic acids, leading to cellular damage and destruction [18,24,295,296]. Plants have evolved several strategies to mitigate Cd toxicity, including chelation, compartmentalization, the reduction of uptake, active efflux, and the enhancement of the antioxidant capacity [18,24]. Among these, changes in antioxidant capacity are often closely related to the concentration of Cd^2+^ available in the plant. Consequently, in many cases, the enhancement of the antioxidant capacity under Cd stress by exogenous substances is primarily due to a reduction in the amount of Cd^2+^ available in the plant [37,38,39,40].

In agricultural practice, our requirements for Cd accumulation and antioxidant capacity are not always aligned. For conventional plants, we seek a combination of a low amount of Cd accumulation and a high antioxidant capacity, termed “low Cd, high antioxidant”, to reduce the risk of Cd entering the food chain. Conversely, for remediation plants, we aim for a high amount of Cd accumulation alongside a strong antioxidant capacity, referred to as “high Cd, high antioxidant”, to enhance the plant’s Cd remediation ability. However, previous research has primarily focused on elucidating Cd accumulation mechanisms, while studies on the regulatory mechanisms of antioxidant capacity and the synergistic effects on resistance and yield are still insufficient. Recent advancements in gene editing technologies, such as CRISPR-Cas9, enable the targeted editing of specific genes. Therefore, a deeper exploration of the regulatory mechanisms underlying plants’ antioxidant capacity and the identification of key genes involved will provide a solid theoretical foundation for developing edible plants with “low Cd, high antioxidant, high yield” traits and remediation plants with “high Cd, high antioxidant, high yield” traits.

## Figures and Tables

**Figure 1 antioxidants-13-01174-f001:**
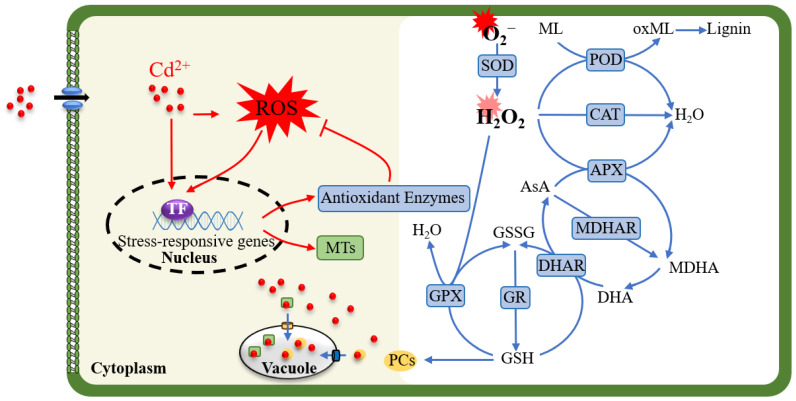
Antioxidant systems involved in the response to Cd stress. Abbreviations: superoxide dismutase, SOD; catalase, CAT; ascorbate peroxidase, APX; peroxidase, POD; guaiacol peroxidase, GPX; glutathione reductase, GR; dehydroascorbate reductase, DHAR; monodehydroascorbate reductase dehydroascorbate, MDHAR; ascorbic acid, AsA; glutathione, GSH; glutathione oxidized, GSSG; phytochelatins, PCs; MTs, metallothioneins; monolignols, ML; oxidized monolignols, oxML; dehydroascorbate, DHA; monodehydroascorbate, MDHA.

**Figure 2 antioxidants-13-01174-f002:**
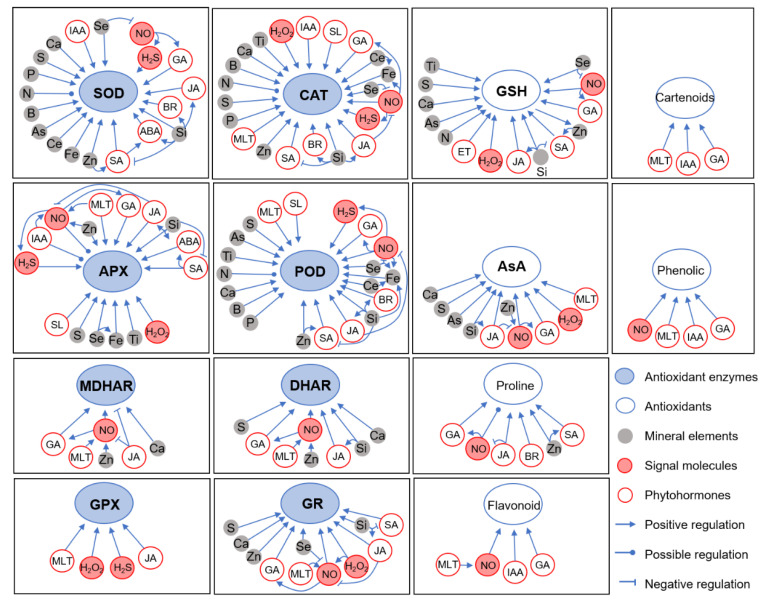
Regulation of antioxidant enzyme activities and antioxidant content by mineral elements, plant growth regulators, and signal molecules.The regulation of the catalytic activity of key antioxidant enzymes involved in induced oxidative stress—such as SOD, CAT, APX, POD, MDHAR, DHAR, GPX, and GR—as well as the levels of major non-enzymatic antioxidants, including GSH, AsA, proline, flavonoids, carotenoids, and phenolics, is influenced by various mineral elements, plant growth regulators, and signaling molecules. Abbreviations: superoxide dismutase, SOD; catalase, CAT; ascorbate peroxidase, APX; peroxidase, POD; guaiacol peroxidase, GPX; glutathione reductase, GR; dehydroascorbate reductase, DHAR; monodehydroascorbate reductase dehydroascorbate, MDHAR; ascorbic acid, AsA; glutathione, GSH; indole-3-acetic acid, IAA; gibberellin, GA; jasmonic acid, JA; brassinosteroids, BRs; abscisic acid, ABA; salicylic acid, SA; ethylene, ETH; strigolactone, SL; melatonin, MLT; hydrogen peroxide, H_2_O_2_; hydrogen sulfide, H_2_S; nitric oxide, NO.

**Figure 3 antioxidants-13-01174-f003:**
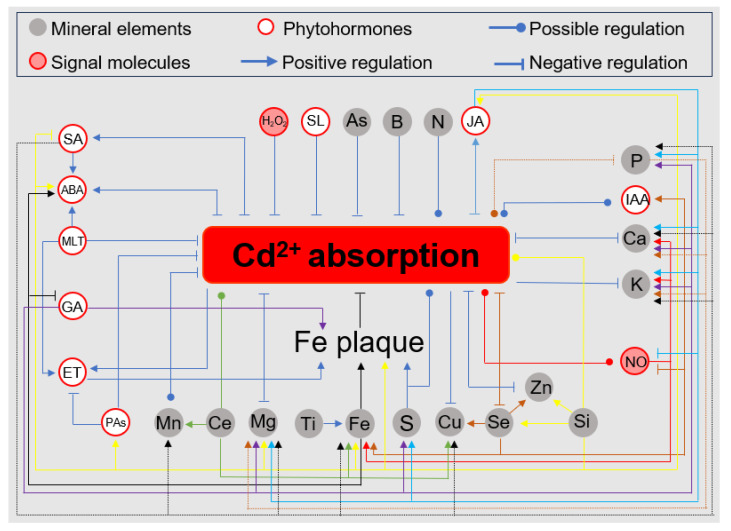
Regulation of Cd^2+^ absorption by various elements, plant growth regulators, and signal molecules. Abbreviations: indole-3-acetic acid, IAA; gibberellin, GA; jasmonic acid, JA; brassinosteroids, BRs; abscisic acid, ABA; salicylic acid, SA; ethylene, ET; strigolactone, SL; melatonin, MLT; polyamines, PAs; hydrogen peroxide, H_2_O_2_; nitric oxide, NO.

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
