# Peer review of "An Overview of the Mechanisms through Which Plants Regulate ROS Homeostasis under Cadmium Stress"

_antioxidants, 2024, doi:10.3390/antiox13101174_

Round 1

Reviewer 1 Report

The main purpose of the article is to review the mechanisms of regulation of cadmium-generated oxidative stress in plants. Topic is relevant because it reviews the current knowledge on the mechanisms of oxidative regulation of plants in cadmium-contaminated areas.  Since this is a review article, it is not a novel article but an updated review of the state of the art. I believe the review article is complete and up to date based on the relevant literature on the topic. The figures presented are somewhat blurred but are informative and correct. The article is suitable for printing in the current version. There are some errata that should be corrected before publication.

In Figure 1. GPX transforms H2O2 into H2O, not H2O2. It would be necessary to correct this erratum in the figure.

It is recommended that higher resolution images be included in the final version of the article, as the figures presented are slightly blurred.

Author Response

Major comments:

The article is suitable for printing in the current version. There are some errata that should be corrected before publication.

We are grateful to the reviewer for your approval of our manuscript.

Detail comments:

In Figure 1. GPX transforms H2Ointo H2O, not H2O2. It would be necessary to correct this erratum in the figure.

We apologize for our error, we have changed ' H2O2' to ' H2O'.

It is recommended that higher resolution images be included in the final version of the article, as the figures presented are slightly blurred.

Done as suggested.

Reviewer 2 Report

In this manuscript, the authors summarizes the positive functions of ROS homeostasis in plants responses to cadmium stress and gives a clear rationale for studying plant response to cadmium stress. The mention of various elements, phytohormones, and signaling molecules involved in cadmium stress response adds depth to the introduction. This work can be improved by including and discussing some of the following topics.

1.       Please include the specific elements, phytohormones, and signaling molecules regulate into this sentence “this review will systematically summarize how various elements, phytohormones, and signaling molecules regulate the antioxidant system under Cd stress and discuss the mechanisms of exogenous regulators that can be employed to enhance antioxidant capacity and mitigate Cd toxicity”.

2.       Line 70, Please check the tense of the “involved”. The same problem needs to be noted in figure 1.

3.       In figure 1, please correct the text to the Times New Roman, and improve the clarity of the picture. Additionally, you can try to beautify the picture and make it richer.

4.       The manuscript repeatedly employs terms like “often” without offering substantial information and evidence to back these claims.

5.       There were a number of review papers available explaining the same concept regarding antioxidant and cadmium stress. Please try to explain the novelty of this paper.

6.       The future perspective should be written more detailed in concluding remarks.

7.       The application of amendments for mitigating oxidative stress is mentioned. So, please briefly describe the usages of CA, BABA, Nanoenzymes, EPS and NAC.

In this manuscript, the authors summarizes the positive functions of ROS homeostasis in plants responses to cadmium stress and gives a clear rationale for studying plant response to cadmium stress. The mention of various elements, phytohormones, and signaling molecules involved in cadmium stress response adds depth to the introduction. This work can be improved by including and discussing some of the following topics.

1.       Please include the specific elements, phytohormones, and signaling molecules regulate into this sentence “this review will systematically summarize how various elements, phytohormones, and signaling molecules regulate the antioxidant system under Cd stress and discuss the mechanisms of exogenous regulators that can be employed to enhance antioxidant capacity and mitigate Cd toxicity”.

2.       Line 70, Please check the tense of the “involved”. The same problem needs to be noted in figure 1.

3.       In figure 1, please correct the text to the Times New Roman, and improve the clarity of the picture. Additionally, you can try to beautify the picture and make it richer.

4.       The manuscript repeatedly employs terms like “often” without offering substantial information and evidence to back these claims.

5.       There were a number of review papers available explaining the same concept regarding antioxidant and cadmium stress. Please try to explain the novelty of this paper.

6.       The future perspective should be written more detailed in concluding remarks.

7.       The application of amendments for mitigating oxidative stress is mentioned. So, please briefly describe the usages of CA, BABA, Nanoenzymes, EPS and NAC.

Author Response

Major comments/ Detail comments:

In this manuscript, the authors summarizes the positive functions of ROS homeostasis in plants responses to cadmium stress and gives a clear rationale for studying plant response to cadmium stress. The mention of various elements, phytohormones, and signaling molecules involved in cadmium stress response adds depth to the introduction. This work can be improved by including and discussing some of the following topics.

  1. Please include the specific elements, phytohormones, and signaling molecules regulate into this sentence “this review will systematically summarize how various elements, phytohormones, and signaling molecules regulate the antioxidant system under Cd stress and discuss the mechanisms of exogenous regulators that can be employed to enhance antioxidant capacity and mitigate Cd toxicity”.

We thank the reviewer for this suggestion, and we have added more details, which are now described as follows: “This review systematically summarizes how various elements, including nitrogen, phosphorus, calcium, iron, and zinc, as well as phytohormones such as abscisic acid, auxin, brassinosteroids, and ethylene, and signaling molecules like nitric oxide, hydrogen peroxide, and hydrogen sulfide, regulate the antioxidant system under cadmium (Cd) stress. Furthermore, it explores the mechanisms by which exogenous regulators can enhance antioxidant capacity and mitigate Cd toxicity.”

  1. Line 70, Please check the tense of the “involved”. The same problem needs to be noted in figure 1.

Done as suggested.

  1. In figure 1, please correct the text to the Times New Roman, and improve the clarity of the picture. Additionally, you can try to beautify the picture and make it richer.

Done as suggested. We have added more useful information to the new Figure 1, enhancing its richness and detail.

  1. The manuscript repeatedly employs terms like “often” without offering substantial information and evidence to back these claims.

We apologize for our inappropriate expression. We have carefully reviewed the entire manuscript and made targeted revisions. 

  1. There were a number of review papers available explaining the same concept regarding antioxidant and cadmium stress. Please try to explain the novelty of this paper.

This is an excellent question and one of the reasons why we undertook the writing of this review. In writing this review, we also noticed many articles focusing on Cd stress and ROS. However, we found that these reviews primarily concentrate on specific aspects, such as plant hormones, signaling molecules, or transcriptional regulation, or on particular species, like rice, which is highly susceptible to Cd toxicity. There is a lack of systematic summaries addressing the interactions among various hormones, signaling molecules, and mineral elements. This review aims to provide a comprehensive overview of the various factors that may regulate Cd-induced ROS bursts and to highlight the interactions between different regulatory mechanisms.

  1. The future perspective should be written more detailed in concluding remarks.

Thank you very much for your suggestions. We have added more content to the Concluding Remarks, which described as follows: “In agricultural practice, our requirements for Cd accumulation and antioxidant capacity are not always aligned. For conventional plants, we seek a combination of low Cd accumulation and high antioxidant capacity, termed "low Cd, high antioxidant," to reduce the risk of Cd entering the food chain. Conversely, for remediation plants, we aim for high Cd accumulation alongside strong antioxidant capacity, referred to as "high Cd, high antioxidant," to enhance the plant's Cd remediation ability. However, previous research has primarily focused on elucidating Cd accumulation mechanisms, while studies on the regulatory mechanisms of antioxidant capacity and the synergistic effects on resistance and yield are still insufficient. Recent advancements in gene editing technologies, such as CRISPR-Cas9, enable targeted editing of specific genes. Therefore, a deeper exploration of the regulatory mechanisms underlying plant antioxidant capacity and the identification of key genes involved will provide a solid theoretical foundation for developing edible plants with "low Cd, high antioxidant, high yield" traits and remediation plants with "high Cd, high antioxidant, high yield" traits.

  1. The application of amendments for mitigating oxidative stress is mentioned. So, please briefly describe the usages of CA, BABA, Nanoenzymes, EPS and NAC.

Thank you very much for your suggestion. We have added a new Table 2 to present this information clearly.

Reviewer 3 Report

The manuscript refers to regulation of cadmium induced oxidative stress in plants. The novelty of the work is low, it is a summary of published data typical for review paper, but it is too general. The authors write very generally, so the text can apply to any heavy metal not only cadmium. They provide too few examples, and do not indicate in which plant material the described changes are observed. I can agree that some ideas of the manuscript are valuable, but some parts are repetition of already known truth. My suggestion is to shorten some parts containing  obvious information and to focus on the particular, narrow problem.

Authors should use both common and Latin name of the plant when mentioned the first time, and only common name of the plant in the further text. They should to be more precise. e.g. melathonin is not a hormone in plants. There are a lot of such  simplifications in the manuscript.

What is the main question addressed by the research? - the review of the antioxidant system in Cd stress 
• Do you consider the topic original or relevant to the field? Does it
address a specific gap in the field? Please also explain why this is/ is not
the case. - The topic i relevant to the field but the manuscript is of low nowelty. It is repetition of some other review papers- a few of new papers referrint to the topic were not cited by the authors
• What does it add to the subject area compared with other published
material? - this is one more review on the topic.
• What specific improvements should the authors consider regarding the
methodology? What further controls should be considered? - does not refer - it is review paper
• Are the conclusions consistent with the evidence and arguments presented
and do they address the main question posed? Please also explain why this
is/is not the case. Yes it is
• Are the references appropriate? No, as I have mention, some new should be included
• Any additional comments on the tables and figures. My comments to fig are included in the pdf version of the manuscript.

Detailed comments are included in the pdf vesion of the manuscript.

Author Response

Major comments:

The manuscript refers to regulation of cadmium induced oxidative stress in plants. The novelty of the work is low, it is a summary of published data typical for review paper, but it is too general. The authors write very generally, so the text can apply to any heavy metal not only cadmium. They provide too few examples, and do not indicate in which plant material the described changes are observed. I can agree that some ideas of the manuscript are valuable, but some parts are repetition of already known truth. My suggestion is to shorten some parts containing  obvious information and to focus on the particular, narrow problem.

Authors should use both common and Latin name of the plant when mentioned the first time, and only common name of the plant in the further text. They should to be more precise. e.g. melathonin is not a hormone in plants. There are a lot of such  simplifications in the manuscript.

What is the main question addressed by the research? - the review of the antioxidant system in Cd stress 
• Do you consider the topic original or relevant to the field? Does it
address a specific gap in the field? Please also explain why this is/ is not
the case. - The topic i relevant to the field but the manuscript is of low nowelty. It is repetition of some other review papers- a few of new papers referrint to the topic were not cited by the authors
• What does it add to the subject area compared with other published
material? - this is one more review on the topic.
• What specific improvements should the authors consider regarding the
methodology? What further controls should be considered? - does not refer - it is review paper
• Are the conclusions consistent with the evidence and arguments presented
and do they address the main question posed? Please also explain why this
is/is not the case. Yes it is
• Are the references appropriate? No, as I have mention, some new should be included
• Any additional comments on the tables and figures. My comments to fig are included in the pdf version of the manuscript.

We greatly appreciate your detailed and insightful suggestions, which have significantly enhanced the appeal and readability of our manuscript. We have made every effort to revise and improve the overall quality of the manuscript in accordance with your suggestion, and we hope this revised version meets your expectations. Our modifications primarily include the following aspects:

1) The common names and Latin names of all species mentioned in the manuscript have been included;

2) Additional descriptions have been added for each Figure, which described as follows:

Description for Figure 1: “When Cd2+ enter the cell, they can trigger an explosion of ROS. Both Cd2+ and ROS can directly activate the transcription of stress-responsive genes, including those encoding antioxidant enzymes and MTs, through specific transcription factors, thereby mitigating damage caused by cadmium stress. On one hand, the cell promotes the accumulation of MTs and PCs, enhancing the formation of Cd2+-MT and Cd2+-PC complexes, which are subsequently transported to the vacuole. This process reduces Cd2+ accumulation in the cytoplasm and decreases ROS production. On the other hand, the cell employs its antioxidant system, comprising both enzymatic and non-enzymatic components, to degrade excess ROS and alleviate oxidative stress.”

Description for Figure 2: “The regulation of the catalytic activity of key antioxidant enzymes involved in -induced oxidative stress—such as SOD, CAT, APX, POD, MDHAR, DHAR, GPX, and GR—as well as the levels of major non-enzymatic antioxidants, including GSH, AsA, proline, flavonoids, carotenoids, and phenolics, is influenced by various mineral elements, plant growth regulators, and signaling molecules.”

3) Species information has been included in the descriptions of each experimental result;

4) We have added the missing references;

Soni, S.; Jha, A. B.; Dubey, R. S.; Sharma, P., Mitigating cadmium accumulation and toxicity in plants: The promising role of nanoparticles. Sci Total Environ 2024, 912.

Vitelli, V.; Giamborino, A.; Bertolini, A.; Saba, A.; Andreucci, A., Cadmium stress signaling pathways in plants: Molecular responses and mechanisms. Curr Issues Mol Biol 2024, 46, (6), 6052-6068.

Zhang, H. W.; Lu, L. L., Transcription factors involved in plant responses to cadmium-induced oxidative stress. Front Plant Sci 2024, 15.

Di, D. W., L, T. T., Yu, Z.L., Cheng, J., Wang, M., Liu, C. F., Wang, Y., Kronzucker, H. J., Yu, M. and Shi, W., Ammonium mitigates cadmium toxicity by activating the bZIP20-APX2/CATA transcriptional module in rice seedlings in an ABA-dependent manner. J Hazard Mater 2024, 135874.

5) We have expanded the discussion to distinguish it from other published reviews.

In addition, we have redesigned Figure 1 and added new Table 2, which enhances the overall appeal of our manuscript. We hope that our revisions will meet your expectations and enhance your satisfaction with our manuscript.

Round 2

Reviewer 2 Report

The manuscript entitled "An overview of the mechanisms through which plants regulate ROS homeostasis under cadmium stress" the authors discuss the recent advances in our understanding of how different exogenous regulators trigger multiple protective mechanisms such as Cd2+ chelation, vesicle sequestration, regulation of Cd2+ uptake, thereby enhancing antioxidant capacity to make the plants Cd2+ stress tolerant. Overall, the manuscript systematically summarizes the various pathways involved in the response to Cd2+ stress. The authors have revised the recommendations we make, and therefore we agree with its publication.

The manuscript entitled "An overview of the mechanisms through which plants regulate ROS homeostasis under cadmium stress" the authors discuss the recent advances in our understanding of how different exogenous regulators trigger multiple protective mechanisms such as Cd2+ chelation, vesicle sequestration, regulation of Cd2+ uptake, thereby enhancing antioxidant capacity to make the plants Cd2+ stress tolerant. Overall, the manuscript systematically summarizes the various pathways involved in the response to Cd2+ stress. The authors have revised the recommendations we make, and therefore we agree with its publication.

Reviewer 3 Report

The manuscript was improved according to great part of my suggestions.

Salicylic acid is not a classical phytohormone, it is plant growth regulator as other molecules e.g. melatonin. Stigolactones are now included into the group of phytohormones. Please modify in the text.